# Earthformer: Exploring Space-Time Transformers for Earth System Forecasting

**Zhihan Gao**[*]
Hong Kong University of Science and Technology
zhihan.gao@connect.ust.hk

**Xingjian Shi**[†]
Amazon Web Services
xjshi@amazon.com

**Hao Wang**
Rutgers University
hw488@cs.rutgers.edu

**Yi Zhu**
Amazon Web Services
yzaws@amazon.com

**Yuyang Wang**
Amazon Web Services
yuyawang@amazon.com

**Mu Li**
Amazon Web Services
mli@amazon.com

**Dit-Yan Yeung**
Hong Kong University of Science and Technology
dyyeung@cse.ust.hk

## Abstract

Conventionally, Earth system (e.g., weather and climate) forecasting relies on numerical simulation with complex physical models and hence is both expensive in computation and demanding on domain expertise. With the explosive growth of spatiotemporal Earth observation data in the past decade, data-driven models that apply Deep Learning (DL) are demonstrating impressive potential for various Earth system forecasting tasks. The Transformer as an emerging DL architecture, despite its broad success in other domains, has limited adoption in this area. In this paper, we propose *Earthformer*, a space-time Transformer for Earth system forecasting. Earthformer is based on a generic, flexible and efficient space-time attention block, named *Cuboid Attention*. The idea is to decompose the data into cuboids and apply cuboid-level self-attention in parallel. These cuboids are further connected with a collection of global vectors. We conduct experiments on the MovingMNIST dataset and a newly proposed chaotic $N$-body MNIST dataset to verify the effectiveness of cuboid attention and figure out the best design of Earthformer. Experiments on two real-world benchmarks about precipitation nowcasting and El Niño/Southern Oscillation (ENSO) forecasting show that Earthformer achieves state-of-the-art performance.

## 1 Introduction

The Earth is a complex system. Variabilities of the Earth system, ranging from regular events like temperature fluctuation to extreme events like drought, hail storm, and El Niño/Southern Oscillation (ENSO), impact our daily life. Among all the consequences, Earth system variabilities can influence crop yields, delay airlines, cause floods and forest fires. Precise and timely forecasting of these variabilities can help people take necessary precautions to avoid crisis, or better utilize natural resources such as wind and solar energy. Thus, improving forecasting models for Earth variabilities (e.g., weather and climate) has a huge socioeconomic impact. Despite its importance, the operational weather and climate forecasting systems have not fundamentally changed for almost 50 years [34]. These operational models, including the state-of-the-art High Resolution Ensemble Forecast (HREF)

---

[*]Work done while being an intern at Amazon Web Services. [†]Contact person.

36th Conference on Neural Information Processing Systems (NeurIPS 2022).

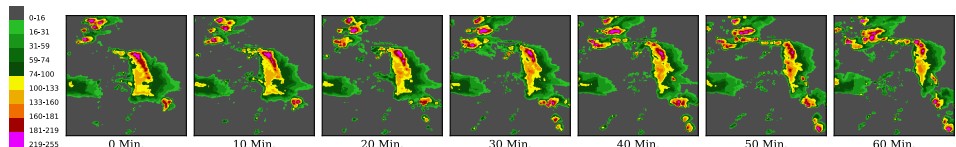

Figure 1: Example Vertically Integrated Liquid (VIL) observation sequence from the Storm EVent ImageRy (SEVIR) dataset. The observation intensity is mapped to pixel value of the range 0-255. The larger value indicates the higher precipitation intensity.

rainfall nowcasting model used in National Oceanic and Atmospheric Administration (NOAA) [32], rely on meticulous numerical simulation of physical models. Such simulation-based systems inevitably fall short in the ability to incorporate signals from newly emerging geophysical observation systems [12], or take advantage of the Petabytes-scale Earth observation data [43].

As an appealing alternative, deep learning (DL) is offering a new approach for Earth system forecasting [34]. Instead of explicitly incorporating physical rules, DL-based forecasting models are trained on the Earth observation data [36]. By learning from a large amount of observations, DL models are able to figure out the system's intrinsic physical rules and generate predictions that outperform simulation-based models [9]. Such technique has demonstrated success in several applications, including precipitation nowcasting [32, 6] and ENSO forecasting [15]. Because the Earth system is chaotic [21], high-dimensional, and spatiotemporal, designing appropriate DL architecture for modeling the system is particularly challenging. Previous works relied on the combination of Recurrent Neural Networks (RNN) and Convolutional Neural Networks (CNN) [36, 37, 43, 13, 45]. These two architectures impose temporal and spatial inductive biases that help capturing spatiotemporal patterns. However, as a chaotic system, variabilities of the Earth system, such as rainfall and ENSO, are highly sensitive to the system's initial conditions and can respond abruptly to internal changes. It is unclear whether the inductive biases in RNN and CNN models still hold for such complex systems.

On the other hand, recent years have witnessed major breakthroughs in DL brought by the wide adoption of Transformer. The model was originally proposed for natural language processing [42, 7], and later has been extended to computer vision [8, 22], multimodal text-image generation [31], graph learning [52], etc. Transformer relies on the attention mechanism to capture data correlations and is powerful at modeling complex and long-range dependencies, both of which appear in Earth systems (See Fig. 1 for an example of Earth observation data). Despite being suitable for the problem, Transformer sees limited adoption for Earth system forecasting. Naively applying the Transformer architecture is infeasible because the $O(N^2)$ attention mechanism is too computationally expensive for the high-dimensional Earth observation data. How to design a space-time Transformer that is good at predicting the future of the Earth systems is largely an open problem to the community.

In this paper, we propose *Earthformer*, a space-time Transformer for Earth system forecasting. To better explore the design of space-time attention, we propose *Cuboid Attention*, which is a generic building block for efficient space-time attention. The idea is to decompose the input tensor to non-overlapping cuboids and apply cuboid-level self-attention in parallel. Since we limit the $O(N^2)$ self-attention inside the local cuboids, the overall complexity is greatly reduced. Different types of correlations can be captured via different cuboid decompositions. By stacking multiple cuboid attention layers with different hyperparameters, we are able to subsume several previously proposed video Transformers [19, 23, 4] as special cases, and also come up with new attention patterns that were not studied before. A limitation of this design is the lack of a mechanism for the local cuboids to communicate with each other. Thus, we introduce a collection of global vectors that attend to all the local cuboids, thereby gathering the overall status of the system. By attending to the global vectors, the local cuboids can grasp the general dynamics of the system and share information with each other.

To verify the effectiveness of cuboid attention and figure out the best design under the Earth system forecasting scenario, we conducted extensive experiments on two synthetic datasets: the MovingMNIST [36] dataset and a newly proposed $N$-body MNIST dataset. Digits in the $N$-body MNIST follow the chaotic 3-body motion pattern [25], which makes the dataset not only more challenging than MovingMNIST but also more relevant to Earth system forecasting. The synthetic experiments reveal the following findings: 1) stacking cuboid attention layers with the Axial attention pattern is both efficient and effective, achieving the best overall performance, 2) adding global vectors provides consistent performance gain without increasing the computational cost, 3) adding hierarchy in the encoder-decoder architecture can improve performance. Based on these findings, we figured out the optimal design for Earthformer and made comparisons with other baselines on the SEVIR [43]

benchmark for precipitation nowcasting and the ICAR-ENSO dataset [15] for ENSO forecasting. Experiments show that Earthformer achieves state-of-the-art (SOTA) performance on both tasks.

## 2 Related Work

**Deep learning architectures for Earth system forecasting.** Conventional DL models for Earth system forecasting are based on CNN and RNN. U-Net with either 2D CNN or 3D CNN have been used for precipitation nowcasting [43], Seasonal Arctic Sea ice prediction [1], and ENSO forecasting [15]. Shi et al. [36] proposed the ConvLSTM network that combines CNN and LSTM for precipitation nowcasting. Wang et al. [45] proposed PredRNN which adds the spatiotemporal memory flow structure to ConvLSTM. To better learn long-term high-level relations, Wang et al. [44] proposed E3D-LSTM that integrates 3D CNN to LSTM. To disentangle PDE dynamics from unknown complementary information, PhyDNet [13] incorporates a new recurrent physical cell to perform PDE-constrained prediction in latent space. Espeholt et al. [9] proposed MetNet-2 that outperforms HREF for forecasting precipitation. The architecture is based on ConvLSTM and dilated CNN. Very recently, there are works that tried to apply Transformer for solving Earth system forecasting problems. Pathak et al. [28] proposed the FourCastNet for global weather forecasting, which is based on Adaptive Fourier Neural Operators (AFNO) [14]. Bai et al. [3] proposed Rainformer for precipitation nowcasting, which is based on an architecture that combines CNN and Swin-Transformer [22]. In our experiments, we can see that Earthformer outperforms Rainformer.

**Space-time Transformers for video modeling.** Inspired by the success of ViT [8] for image classification, space-time Transformer is adopted for improved video understanding. In order to bypass the huge memory consumption brought by joint spatiotemporal attention, several pioneering work proposed efficient alternatives, such as divided attention [4], axial attention [19, 4], factorized encoder [27, 2] and separable attention [54]. Beyond minimal adaptation from ViT, some recent work introduced more prior to the design of space-time transformers, including trajectory [29], multi-scale [23, 11] and multi-view [49]. However, no prior work focuses on exploring the design of space-time Transformers for Earth system forecasting.

**Global and local attention in vision Transformers.** To make self-attention more efficient in terms of both memory consumption and speed, recent works have adapted the essence of CNN to perform local attention in transformers [16, 52]. HaloNets [41] develops a new self-attention model family consisting of simple local self-attention and convolutional hybrids, which outperform both CNN and vanilla ViT on a range of downstream vision tasks. GLiT [5] introduces a locality module and uses neural architecture search to find an efficient backbone. Focal transformer [51] proposes focal self-attention that can incorporate both fine-grained local and coarse-grained global interactions. However, these architectures are not directly applicable to spatiotemporal forecasting. Besides, they are also different from our design because we keep $K$ global vectors to summarize the statistics of the dynamic system and connect the local cuboids; experiments show that such a global vector design is crucial to successful spatiotemporal forecasting.

## 3 Model

Similar to previous works [36, 43, 3], we formulate Earth system forecasting as a spatiotemporal sequence forecasting problem. The Earth observation data, such as radar echo maps from NEXRAD [17] and climate data from CIMP6 [10], are represented as a spatiotemporal sequence $[\mathcal{X}_i]_{i=1}^{T}, \mathcal{X}_i \in \mathbb{R}^{H \times W \times C_{\text{in}}}$. Based on these observations, the model predicts the $K$-step-ahead future $[\mathcal{Y}_{T+i}]_{i=1}^{K}, \mathcal{Y}_{T+i} \in \mathbb{R}^{H \times W \times C_{\text{out}}}$. Here, $H, W$ denote the spatial resolution, and $C_{\text{in}}, C_{\text{out}}$ denote the number of measurements available at each space-time coordinate from the input and target sequences, respectively. As illustrated in Fig. 2, our proposed *Earthformer* is a hierarchical Transformer encoder-decoder based on *Cuboid Attention*. The input observations are encoded as a hierarchy of hidden states and then decoded to the prediction target. In what follows, we will present the detailed design of cuboid attention and the hierarchical encoder-decoder architecture adopted in Earthformer.

### 3.1 Cuboid Attention

Compared with images and text, spatiotemporal data in Earth systems usually have higher dimensionality. As a consequence, applying Transformers to this task is challenging. For example, for a 3D

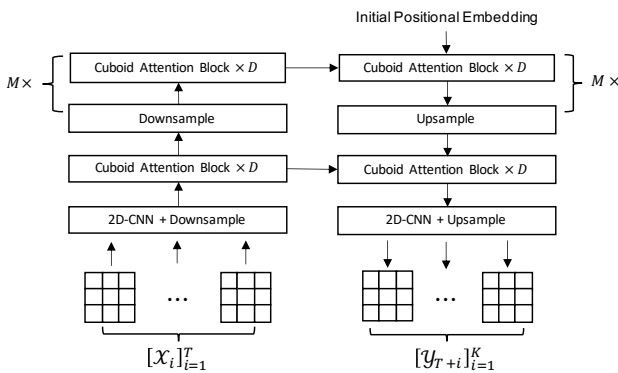

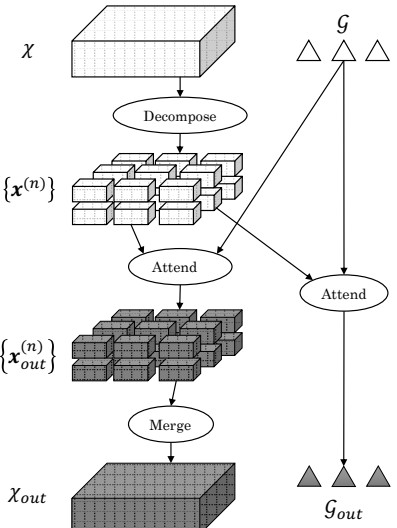

Figure 2: Illustration of the Earthformer architecture. It is a hierarchical Transformer encoder-decoder based on cuboid attention. The input sequence has length $T$ and the target sequence has length $K$. "$\times D$" means to stack $D$ cuboid attention blocks with residual connection. "$M\times$" means to have $M$ layers of hierarchies.

Figure 3: Illustration of the cuboid attention layer with global vectors.

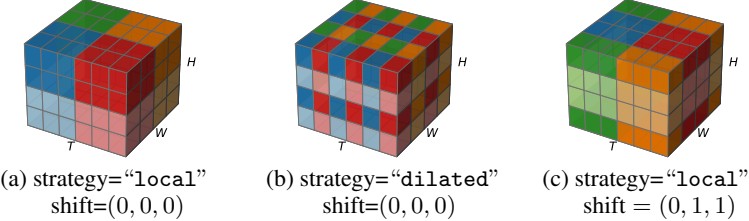

(a) strategy="local" shift=(0,0,0)

(b) strategy="dilated" shift=(0,0,0)

(c) strategy="local" shift $= (0,1,1)$

Figure 4: Illustration of cuboid decomposition strategies when the input shape is $(T, H, W) = (6, 4, 4)$, and cuboid size $(b_T, b_H, b_W) = (3, 2, 2)$. Cells that have the same color belong to the same cuboid and will attend to each other. shift $= (0, 1, 1)$ shifts the cuboid decomposition by 1 pixel along height and width dimensions. strategy = "local" means to aggregate contiguous $(b_T, b_H, b_W)$ pixels as a cuboid. strategy = "dilated" means to aggregate pixels every $\lceil \frac{T}{b_T} \rceil$ ($\lceil \frac{H}{b_H} \rceil$, $\lceil \frac{W}{b_W} \rceil$) steps along time (height, width) dimension. (Best viewed in color).

tensor with shape $(T, H, W)$, the complexity of the vanilla self-attention is $O(T^2 H^2 W^2)$ and can be computationally infeasible. Previous literature proposed various structure-aware space-time attention mechanisms to reduce the complexity [19, 23, 4].

These space-time attention mechanisms share the common design of stacking multiple elementary attention layers that focus on different types of data correlations (e.g., temporal correlation and spatial correlation). Steming from this observation, we propose the generic cuboid attention layer that involves three steps: "decompose", "attend", and "merge".

**Decompose.** We first decompose the input spatiotemporal tensor $\mathcal{X} \in \mathbb{R}^{T \times H \times W \times C}$ into a sequence of cuboids $\{\boldsymbol{x}^{(n)}\}$.

$$\{\boldsymbol{x}^{(n)}\} = \texttt{Decompose}(\mathcal{X}, \text{cuboid\_size}, \text{strategy}, \text{shift}), \tag{1}$$

where cuboid\_size $= (b_T, b_H, b_W)$ is the size of the local cuboid, strategy $\in \{\text{"local"}, \text{"dilated"}\}$ controls whether to adopt the local decomposition strategy or the dilated decomposition strategy [4], shift $= (s_T, s_H, s_W)$ is the window shift offset [22]. Fig. 4 provides three examples showing how an input tensor will be decomposed following different hyperparameters of $\texttt{Decompose}(\cdot)$. There are a total number of $\lceil \frac{T}{b_T} \rceil \lceil \frac{H}{b_H} \rceil \lceil \frac{W}{b_W} \rceil$ cuboids in $\{\boldsymbol{x}^{(n)}\}$. To simplify the notation, we assume that $T, H, W$ are divisible by $b_T, b_H, b_W$. In the implementation, we pad the input tensor if it is not divisible.

Assume $\boldsymbol{x}^{(n)}$ is the $(n_T, n_H, n_W)$-th cuboid in $\{\boldsymbol{x}^{(n)}\}$. The $(i, j, k)$-th element of $\boldsymbol{x}^{(n)}$ can be mapped to the $(i', j', k')$-th element of $\mathcal{X}$ via Eqn. 2 if the strategy is "local" or Eqn. 3 if the strategy is "dilated".

$$
\begin{aligned}
i' &\leftrightarrow s_T + b_T(n_T - 1) + i \mod T & i' &\leftrightarrow s_T + b_T(i - 1) + n_T \mod T \\
j' &\leftrightarrow s_H + b_H(n_H - 1) + j \mod H \quad (2) & j' &\leftrightarrow s_H + b_H(j - 1) + n_H \mod H \quad (3) \\
k' &\leftrightarrow s_W + b_W(n_W - 1) + k \mod W & k' &\leftrightarrow s_W + b_W(k - 1) + n_W \mod W
\end{aligned}
$$

Since the mapping is bijective, one can then map the elements from $\mathcal{X}$ to $\{\boldsymbol{x}^{(n)}\}$ via the inverse operation.

**Attend.** After decomposing the input tensor into a sequence of non-overlapping cuboids $\{\boldsymbol{x}^{(n)}\}$, we apply self-attention within each cuboid in parallel.

$$
\boldsymbol{x}_{\text{out}}^{(n)} = \text{Attention}_\Theta(\boldsymbol{x}^{(n)}, \boldsymbol{x}^{(n)}, \boldsymbol{x}^{(n)}), 1 \le n \le N. \tag{4}
$$

The query, key, and value matrices $\boldsymbol{Q}$, $\boldsymbol{K}$, and $\boldsymbol{V}$ of $\text{Attention}_\theta(\boldsymbol{Q}, \boldsymbol{K}, \boldsymbol{V}) = \text{Softmax}\left((\boldsymbol{W_Q Q})(\boldsymbol{W_K K})^T/\sqrt{C}\right)(\boldsymbol{W_V V})$ are all flattened versions of $\boldsymbol{x}^{(n)}$, and we unravel the resulting matrix back to a 3D tensor. $\boldsymbol{W_Q}$, $\boldsymbol{W_K}$ and $\boldsymbol{W_V}$ are linear projection weights and are abbreviated together as $\Theta$. The self-attention parameter $\Theta$ are shared across all cuboids. The computational complexity of the "attend" step is $O\left(\lceil\frac{T}{b_T}\rceil\lceil\frac{H}{b_H}\rceil\lceil\frac{W}{b_W}\rceil(b_T b_H b_W)^2\right) \approx O(THW \cdot b_T b_H b_W)$, which scales linearly with the cuboid size. Since the cuboid size can be much smaller than the size of the input tensor, the layer is more efficient than full attention.

**Merge.** $\text{Merge}(\cdot)$ is the inverse operation of $\text{Decompose}(\cdot)$. The sequence of cuboids obtained after the attention step $\{\boldsymbol{x}_{\text{out}}^{(n)}\}$ are merged back to the original input shape to produce the final output of cuboid attention, as shown in Eqn. 5. The mapping follows the same bijections in Eqn. 2 and Eqn. 3.

$$
\mathcal{X}_{\text{out}} = \text{Merge}(\{\boldsymbol{x}_{\text{out}}^{(n)}\}_n, \text{cuboid\_size}, \text{strategy}, \text{shift}). \tag{5}
$$

We combine the "decompose", "attend" and "merge" steps described in Eqn. 1,4,5 to construct the generic cuboid attention as in Eqn. 6.

$$
\mathcal{X}_{\text{out}} = \text{CubAttn}_\Theta(\mathcal{X}, \text{cuboid\_size}, \text{strategy}, \text{shift}). \tag{6}
$$

**Explore cuboid attention patterns.** By stacking multiple cuboid attention layers with different choices of "cuboid_size", "strategy" and "shift", we are able to efficiently explore existing and potentially more effective space-time attention. In this paper, we explore the cuboid attention patterns as listed in Table 1. From the table, we can see that cuboid attention subsumes previously proposed space-time attention methods like axial attention, video swin-Transformer, and divided space-time attention. Also, we manually picked the patterns that are reasonable and not computationally expensive as our search space. The flexibility of cuboid attention allows us to conduct Neural Architecture Search (NAS) to automatically search for a pattern but we will leave it as future work.

## 3.2 Global Vectors

One limitation of the previous formulation is that the cuboids do not communicate with each other. This is undesirable because each cuboid is not capable of understanding the global dynamics of the system. Thus, inspired by the [CLS] token adopted in BERT [7, 53], we propose to introduce a collection of $P$ global vectors $\mathcal{G} \in \mathbb{R}^{P \times C}$ to help cuboids scatter and gather crucial global information. When each cuboid is performing the self-attention, the elements will not only attend to the other elements within the same cuboid but also attend to the global vectors $\mathcal{G}$. We revise Eqn. 4 to Eqn. 7 to enable local-global information exchange. We also use Eqn. 8 to update the global vectors $\mathcal{G}$ by aggregating the information from all elements of the input tensor $\mathcal{X}$.

$$
\boldsymbol{x}_{\text{out}}^{(n)} = \text{Attention}_\Theta\left(\boldsymbol{x}^{(n)}, \text{Cat}(\boldsymbol{x}^{(n)}, \mathcal{G}), \text{Cat}(\boldsymbol{x}^{(n)}, \mathcal{G})\right), 1 \le n \le N. \tag{7}
$$

$$
\mathcal{G}_{\text{out}} = \text{Attention}_\Phi\left(\mathcal{G}, \text{Cat}(\mathcal{G}, \mathcal{X}), \text{Cat}(\mathcal{G}, \mathcal{X})\right). \tag{8}
$$

Table 1: Configurations of the cuboid attention patterns explored in the paper. The input tensor has shape $(T, H, W)$. If "shift" or "strategy" is not given, we use shift $= (0, 0, 0)$ and strategy $=$ "local" by default. When stacking multiple cuboid attention layers, each layer will be coupled with layer normalization layers and feed-forward network as in the Pre-LN Transformer [48]. The first row shows the configuration of the generic cuboid attention.

| Name | Configurations | Values |
|---|---|---|
| Generic Cuboid Attention | cuboid_size
shift
strategy | $(\mathtt{T_1, H_1, W_1}) \rightarrow (\mathtt{T_2, H_2, W_2}) \rightarrow \cdots \rightarrow (\mathtt{T_L, H_L, W_L})$
$(\mathtt{P_1, M_1, M_1}) \rightarrow (\mathtt{P_2, M_2, M_2}) \rightarrow \cdots \rightarrow (\mathtt{P_L, M_L, M_L})$
"loc./dil." $\rightarrow$ "loc./dil." $\rightarrow \cdots \rightarrow$ "loc./dil." |
| Axial | cuboid_size | $(\mathtt{T, 1, 1}) \rightarrow (\mathtt{1, H, 1}) \rightarrow (\mathtt{1, 1, W})$ |
| Divided Space-Time | cuboid_size | $(\mathtt{T, 1, 1}) \rightarrow (\mathtt{1, H, W})$ |
| Video-Swin $P \times M$ | cuboid_size
shift | $(\mathtt{P, M, M}) \rightarrow (\ \mathtt{P}\ ,\ \mathtt{M}\ ,\ \mathtt{M}\ )$
$(\mathtt{0, 0, 0}) \rightarrow (\mathtt{P/2, M/2, M/2})$ |
| Spatial Local-Dilate-$M$ | cuboid_size
strategy | $(\mathtt{T, 1, 1}) \rightarrow (\mathtt{1, M, M}) \rightarrow (\mathtt{1, M, M})$
"local" $\rightarrow$ "local" $\rightarrow$ "dilated" |
| Axial Space Dilate-$M$ | cuboid_size
strategy | $(\mathtt{T, 1, 1}) \rightarrow (\mathtt{1, H/M, 1}) \rightarrow (\mathtt{1, H/M, 1}) \rightarrow (\mathtt{1, 1, W/M}) \rightarrow (\mathtt{1, 1, W/M})$
"local" $\rightarrow$ "dilated" $\rightarrow$ "local" $\rightarrow$ "dilated" $\rightarrow$ "local" |

Here, $\mathtt{Cat}(\cdot)$ flattens and concatenates its input tensors. By combining Eqn. 1,7,8,5, we abbreviate the overall computation of the cuboid attention layer with global vectors as in Eqn. 9.

$$\mathcal{X}_{\text{out}} = \mathtt{CubAttn}_\Theta(\mathcal{X}, \mathcal{G}, \text{cuboid\_size}, \text{strategy}, \text{shift}),$$
$$\mathcal{G}_{\text{out}} = \mathtt{Attn}_\Phi^{\text{global}}(\mathcal{G}, \mathcal{X}).$$
$$\text{(9)}$$

The additional complexity caused by the global vectors is approximately $O\left(THW \cdot P + P^2\right)$. Given that $P$ is usually small (in our experiments, $P$ is at most 8), the computational overhead induced by the global structure is negligible. The architecture of the cuboid attention layer is illustrated in Fig. 3.

### 3.3 Hierarchical Encoder-Decoder Architecture

Earthformer adopts a hierarchical encoder-decoder architecture illustrated in Fig. 2. The hierarchical architecture gradually encodes the input sequence to multiple levels of representations and generates the prediction via a coarse-to-fine procedure. Each hierarchy stacks $D$ cuboid attention blocks. The cuboid attention block in the encoder uses one of the patterns described in Table 1, and each cuboid block in the decoder adopts the "Axial" pattern. To reduce the spatial resolution of the input to cuboid attention layers, we include a pair of initial downsampling and upsampling modules that consist of stacked 2D-CNN and Nearest Neighbor Interpolation (NNI) layers. Different from other papers that adopt Transformer for video prediction [19, 30], we generate the predictions in a non-auto-regressive fashion rather than an auto-regressive patch-by-patch fashion. This means that our decoder directly generates the predictions from the initial learned positional embeddings. We also conducted experiments with an auto-regressive decoder based on visual codebook [33]. However, the auto-regressive decoder underperforms the non-auto-regressive decoder in terms of forecasting skill scores. The comparison between non-auto-regressive decoder and auto-regressive decoder is shown in Appendix C.

## 4 Experiments

We first conducted experiments on two synthetic datasets, MovingMNIST and a newly proposed $N$-body MNIST, to verify the effectiveness of Earthformer and conduct ablation study on our design choices. Results on these two datasets lead to the following findings: 1) Among all patterns listed in Table 1, "Axial" achieves the best overall performance; 2) Global vectors bring consistent performance gain with negligible increase in computational cost; 3) Using a hierarchical coarse-to-fine structure can boost the performance. Based on these findings, we figured out the optimal design of Earthformer and compared it with other state-of-the-art models on two real-world datasets: SEVIR [43] and ICAR-ENSO[2]. On both datasets, Earthformer achieved the best overall performance. The statistics of all the datasets used in the experiments are shown in Table 2. We normalized the data to the range $[0, 1]$ and trained all the models with the Mean-Squared Error (MSE) loss. More implementation details are shown in Appendix A.

---

[2]Dataset available at https://tianchi.aliyun.com/dataset/dataDetail?dataId=98942

Table 2: Statistics of the datasets used in the experiments.

| Dataset | Size | | | Seq. Len. | | Spatial Resolution |
|---|---|---|---|---|---|---|
| | train | val | test | in | out | $H \times W$ |
| MovingMNIST | 8,100 | 900 | 1,000 | 10 | 10 | $64 \times 64$ |
| $N$-body MNIST | 20,000 | 1,000 | 1,000 | 10 | 10 | $64 \times 64$ |
| SEVIR | 35,718 | 9,060 | 12,159 | 13 | 12 | $384 \times 384$ |
| ICAR-ENSO | 5,205 | 334 | 1,667 | 12 | 14 | $24 \times 48$ |

Table 3: Ablation study on the importance of adopting a hierarchical encoder-decoder. We conducted experiments on MovingMNIST. "Depth $D$" means the model stacks $D$ cuboid attention blocks and there is no hierarchical structure. "Depth $D1, D2$" means the model stacks $D1$ cuboid attention blocks, applies the pooling layer, and stacks another $D2$ cuboid attention blocks.

| Model | #Param. (M) | GFLOPS | Metrics | | |
|---|---|---|---|---|---|
| | | | MSE ↓ | MAE ↓ | SSIM ↑ |
| Depth 2 | 1.4 | 17.9 | 63.80 | 140.6 | 0.8324 |
| Depth 4 | 3.1 | 36.3 | 52.46 | 114.8 | 0.8685 |
| Depth 6 | 4.9 | 54.6 | 50.49 | 110.0 | 0.8738 |
| Depth 8 | 6.6 | 73.0 | 48.04 | 104.6 | 0.8797 |
| Depth 1,1 | 1.4 | 11.5 | 60.99 | 135.7 | 0.8388 |
| Depth 2,2 | 3.1 | 18.9 | 50.41 | 106.9 | 0.8805 |
| Depth 3,3 | 4.9 | 26.3 | 47.69 | **100.1** | **0.8873** |
| Depth 4,4 | 6.6 | 33.7 | **46.91** | 101.5 | 0.8825 |

## 4.1 Experiments on Synthetic Datasets

**MovingMNIST.** We follow [38] to use the public MovingMNIST dataset[3]. The dataset contains 10,000 sequences. Each sequence shows 2 digits moving inside a $64 \times 64$ frame. We split the dataset to use 8,100 samples for training, 900 samples for validation and 1,000 samples for testing. The task is to predict the future 10 frames for each sequence conditioned on the first 10 frames.

**$N$-body MNIST.** The Earth is a complex system in which an extremely large number of variables interact with each other. Compared with the Earth system, the dynamics of the synthetic MovingMNIST dataset, in which the digits move independently with constant speed, is over-simplified. Thus, achieving good performance on MovingMNIST does not imply that the model is capable of modeling complex interactions in the Earth system. On the other hand, the real-world Earth observation data, though experiencing rapid development, are still noisy and may not provide useful insights for model development. Therefore, we extend MovingMNIST to $N$-body MNIST, where $N$ digits are moving with the $N$-body motion pattern inside a $64 \times 64$ frame. Each digit has its mass and is subjected to the gravity from other digits. We choose $N = 3$ in the experiments so that the digits will follow the chaotic 3-body motion [25]. The highly non-linear interactions in $N$-body MNIST make it much more challenging than the original MovingMNIST. We generate 20,000 sequences for training, 1,000 for validation and 1,000 for testing. Perceptual examples of the dataset can be found at the first two rows of Fig. 5. In Appendix D, we demonstrate the chaotic behavior of $N$-body MNIST.

**Hierarchical v.s. non-hierarchical.** We choose "Axial" without global vectors as our cuboid attention pattern and compare the performance of non-hierarchical and hierarchical architectures on MovingMNIST. The ablation study on the importance of adopting a hierarchical encoder-decoder is shown in Table 3. We can see that the hierarchical architecture has similar FLOPS with the non-hierarchical architectures while being better in MSE. This observation is consistent as we increase the depth until the performance saturates.

**Cuboid pattern search.** The design of cuboid attention greatly facilitates the search for optimal space-time attention. We compare the patterns listed in Table 1 on both MovingMNIST and $N$-body MNIST to investigate the effectiveness and efficiency of different space-time attention methods on spatiotemporal forecasting tasks. Besides the previously proposed space-time attention methods, we also include new configurations that are reasonable and not computationally expensive in our search space. For each pattern, we also compare the variant that uses global vectors. Results are summarized in Table 4. We find that the "Axial" pattern is both effective and efficient and adding global vectors improves performance for all patterns while having similar FLOPS. We thus pick "Axial + global" as the pattern in Earthformer when conducting experiments on real-world datasets.

---

[3]MovingMNIST: https://github.com/mansimov/unsupervised-videos

Table 4: Ablation study of different cuboid attention patterns and the effect of global vectors on MovingMNIST and $N$-body MNIST. The variant that achieved the best performance is in boldface while the second best is underscored. We also compared the performance of the cuboid attention patterns with and without global vectors and highlight the better one with grey background.

| Model | #Param. (M) | GFLOPS | MovingMNIST | | | $N$-body MNIST | | |
|---|---|---|---|---|---|---|---|---|
| | | | MSE ↓ | MAE ↓ | SSIM ↑ | MSE ↓ | MAE ↓ | SSIM ↑ |
| Axial | 6.61 | 33.7 | 46.91 | 101.5 | 0.8825 | 15.89 | 41.38 | 0.9510 |
| + global ★ | 7.61 | 34.0 | **41.79** | **92.78** | **0.8961** | **14.82** | **39.93** | **0.9538** |
| DST | 5.70 | 35.2 | 57.43 | 118.6 | 0.8623 | 18.24 | 45.88 | 0.9435 |
| + global | 6.37 | 35.5 | 52.92 | 108.3 | 0.8760 | 17.77 | 45.84 | 0.9433 |
| Video Swin 2x8 | 5.66 | 31.1 | 54.45 | 111.7 | 0.8715 | 19.89 | 49.02 | 0.9374 |
| + global | 6.33 | 31.4 | 52.70 | 108.5 | 0.8766 | 19.53 | 48.43 | 0.9389 |
| Video Swin 10x8 | 5.89 | 39.2 | 63.34 | 125.3 | 0.8525 | 23.35 | 53.17 | 0.9274 |
| + global | 6.56 | 39.4 | 62.15 | 123.4 | 0.8541 | 22.81 | 52.94 | 0.9293 |
| Spatial Local-Global 2 | 6.61 | 33.3 | 59.88 | 122.1 | 0.8572 | 23.24 | 54.63 | 0.9263 |
| + global | 7.61 | 33.7 | 59.42 | 122.9 | 0.8565 | 21.88 | 52.49 | 0.9305 |
| Spatial Local-Global 4 | 6.61 | 33.5 | 58.72 | 118.5 | 0.8600 | 21.02 | 49.82 | 0.9344 |
| + global | 7.61 | 33.9 | 54.84 | 115.5 | 0.8585 | 19.82 | 48.12 | 0.9371 |
| Axial Space Dilate 2 | 8.59 | 41.8 | 50.11 | 104.4 | 0.8814 | 15.97 | 42.19 | 0.9494 |
| + global | 10.30 | 42.4 | 46.86 | 98.95 | 0.8884 | 15.73 | 41.85 | 0.9510 |
| Axial Space Dilate 4 | 8.59 | 41.6 | 47.40 | 99.31 | 0.8865 | 19.49 | 51.04 | 0.9352 |
| + global | 10.30 | 42.2 | 45.11 | 95.98 | 0.8928 | 17.91 | 46.35 | 0.9440 |

Table 5: Comparison of Earthformer with baselines on MovingMNIST and $N$-body MNIST.

| Model | #Param. (M) | GFLOPS | MovingMNIST | | | $N$-body MNIST | | |
|---|---|---|---|---|---|---|---|---|
| | | | MSE ↓ | MAE ↓ | SSIM ↑ | MSE ↓ | MAE ↓ | SSIM ↑ |
| UNet [43] | 16.6 | 0.9 | 110.4 | 249.4 | 0.6170 | 38.90 | 94.29 | 0.8260 |
| ConvLSTM [36] | 14.0 | 30.1 | 62.04 | 126.9 | 0.8477 | 32.15 | 72.64 | 0.8886 |
| PredRNN [45] | 23.8 | 232.0 | 52.07 | 108.9 | 0.8831 | 21.76 | 54.32 | 0.9288 |
| PhyDNet [13] | 3.1 | 15.3 | 58.70 | 124.1 | 0.8350 | 28.97 | 78.66 | 0.8206 |
| E3D-LSTM [44] | 12.9 | 302.0 | 55.31 | 101.6 | 0.8821 | 22.98 | 62.52 | 0.9131 |
| Rainformer [3] | 19.2 | 1.2 | 85.83 | 189.2 | 0.7301 | 38.89 | 96.47 | 0.8036 |
| Earthformer w/o global | 6.6 | 33.7 | 46.91 | 101.5 | 0.8825 | 15.89 | 41.38 | 0.9510 |
| Earthformer | 7.6 | 34.0 | **41.79** | **92.78** | **0.8961** | **14.82** | **39.93** | **0.9538** |

**Comparison to the state of the art.** We evaluate six spatiotemporal forecasting algorithms: UNet [43], ConvLSTM [36], PredRNN [45], PhyDNet [13], E3D-LSTM [44] and Rainformer [3]. The results are in Table 5. Note that the MovingMNIST performance on several papers [13] is obtained by training the model with on-the-fly generated digits while we pre-generate the digits and train all models on a fixed dataset. Comparing the numbers in the table with numbers shown in these papers are not fair. We train all baselines from scratch on both MovingMNIST and $N$-body MNIST using the default hyperparameters and configurations in their officially released code[4].

**Qualitative results on $N$-body MNIST.** Fig. 5 shows the generation results of different methods on a sample sequence from the $N$-body MNIST test set. The qualitative example demonstrates that our Earthformer is capable of learning long-range interactions among digits and correctly predicting their future motion trajectories. Also, we can see that Earthformer is able to more accurately predict the position of the digits with the help of global vectors. On the contrary, none of the baseline algorithms that achieved solid performance on MovingMNIST gives the correct and precise position of the digit "0" in the last frame. They either predict incorrect motion trajectories (PredRNN and E3D-LSTM), or generate highly blurry predictions (Rainformer, UNet and PhyDNet) to accommodate the uncertainty about the future.

## 4.2 SEVIR Precipitation Nowcasting

Storm EVent ImageRy (SEVIR) [43] is a spatiotemporally aligned dataset containing over 10,000 weather events. Each event consists of $384 \text{ km} \times 384 \text{ km}$ image sequences spanning over 4 hours. Images in SEVIR were sampled and aligned across five different data types: three channels (C02,

---

[4]Except for Rainformer which originally has 212M parameters and thus suffers from overfitting severely.

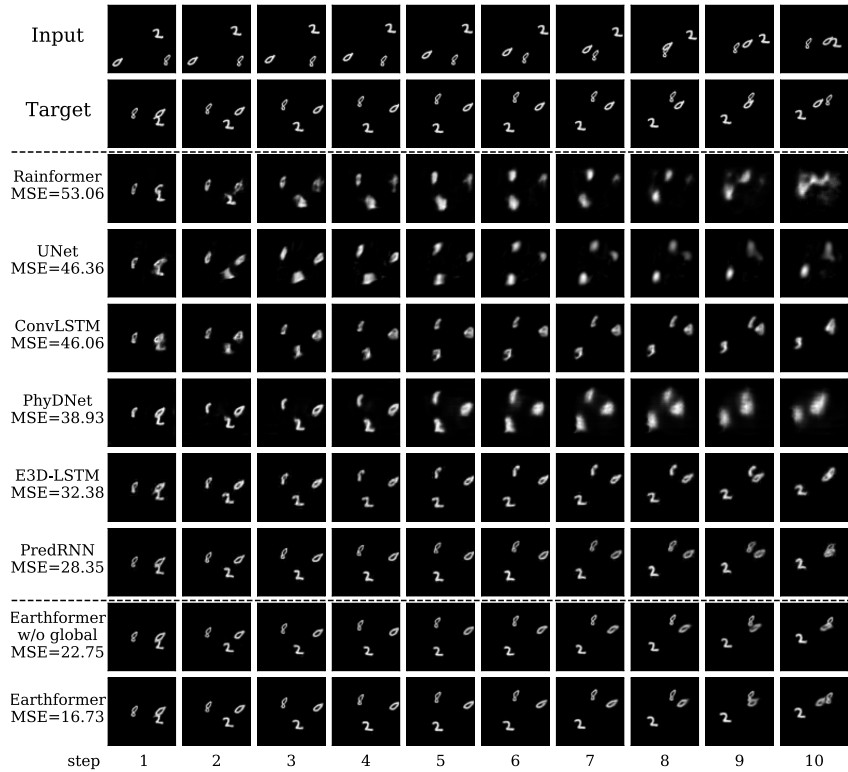

Figure 5: A set of examples showing the perceptual quality of the predictions on the $N$-body MNIST test set. From top to bottom: input frames, target frames, predictions by Rainformer [3], UNet [43], ConvLSTM [36], PhyDNet [13], E3D-LSTM [44], PredRNN [45], Earthformer without using global vectors, Earthformer. The results are sorted according to the MSE.

Table 6: Performance comparison on SEVIR. We include `Critical Success Index` (CSI) besides MSE as evaluation metrics. The `CSI`, a.k.a intersection over union (IOU), is calculated at different precipitation thresholds and denoted as `CSI-thresh`.

| Model | #Param. (M) | GFLOPS | Metrics | | | | | | | |
|---|---|---|---|---|---|---|---|---|---|---|
| | | | CSI-M↑ | CSI-219↑ | CSI-181↑ | CSI-160↑ | CSI-133↑ | CSI-74↑ | CSI-16↑ | MSE ($10^{-3}$)↓ |
| Persistence | - | - | 0.2613 | 0.0526 | 0.0969 | 0.1278 | 0.2155 | 0.4705 | 0.6047 | 11.5338 |
| UNet [43] | 16.6 | 33 | 0.3593 | 0.0577 | 0.1580 | 0.2157 | 0.3274 | 0.6531 | 0.7441 | 4.1119 |
| ConvLSTM [36] | 14.0 | 527 | 0.4185 | 0.1288 | 0.2482 | 0.2928 | 0.4052 | 0.6793 | 0.7569 | 3.7532 |
| PredRNN [45] | 46.6 | 328 | 0.4080 | 0.1312 | 0.2324 | 0.2767 | 0.3858 | 0.6713 | 0.7507 | 3.9014 |
| PhyDNet [13] | 13.7 | 701 | 0.3940 | 0.1288 | 0.2309 | 0.2708 | 0.3720 | 0.6556 | 0.7059 | 4.8165 |
| E3D-LSTM [44] | 35.6 | 523 | 0.4038 | 0.1239 | 0.2270 | 0.2675 | 0.3825 | 0.6645 | 0.7573 | 4.1702 |
| Rainformer [3] | 184.0 | 170 | 0.3661 | 0.0831 | 0.1670 | 0.2167 | 0.3438 | 0.6585 | 0.7277 | 4.0272 |
| Earthformer w/o global | 13.1 | 257 | 0.4356 | 0.1572 | 0.2716 | 0.3138 | 0.4214 | 0.6859 | 0.7637 | 3.7002 |
| Earthformer | 15.1 | 257 | **0.4419** | **0.1791** | **0.2848** | **0.3232** | **0.4271** | **0.6860** | 0.7513 | **3.6957** |

C09, C13) from the GOES-16 advanced baseline imager, NEXRAD Vertically Integrated Liquid (VIL) mosaics, and GOES-16 Geostationary Lightning Mapper (GLM) flashes. The SEVIR benchmark supports scientific research on multiple meteorological applications including precipitation nowcasting, synthetic radar generation, front detection, etc. We adopt SEVIR for benchmarking precipitation nowcasting, i.e., to predict the future VIL up to 60 minutes (12 frames) given 65 minutes context VIL (13 frames). Fig. 1 shows an example of VIL observation sequences in SEVIR.

Besides MSE, we also include the `Critical Success Index` (CSI), which is commonly used in precipitation nowcasting and is defined as $\mathtt{CSI} = \frac{\#\mathtt{Hits}}{\#\mathtt{Hits}+\#\mathtt{Misses}+\#\mathtt{F.Alarms}}$. To count the #Hits (truth=1, pred=1), #Misses (truth=1, pred=0) and #F.Alarms (truth=0, pred=1), the prediction and the ground-truth are rescaled back to the range 0-255 and binarized at thresholds $[16, 74, 133, 160, 181, 219]$. We report `CSI` at different thresholds and also their mean `CSI-M`.

SEVIR is much larger than MovingMNIST and $N$-body MNIST and has higher resolution. We thus slightly adjust the configurations of baselines based on those for MovingMNIST for fair comparison. Detailed configurations are shown in Appendix A. The experiment results are listed in Table 6. Earthformer consistently outperforms baselines on almost all metrics and brings significant performance gain especially at high thresholds like `CSI-219`, which are more valued by the communities.

Table 7: Performance comparison on ICAR-ENSO. $C$-Nino3.4-M and $C$-Nino3.4-WM are the mean and the weighted mean of the correlation skill $C^{\text{Nino3.4}}$ over $K = 12$ forecasting steps. $C$-Nino3.4-WM assigns more weights to longer-term prediction scores. MSE is calculated between the spatiotemporal SST anomalies prediction and the corresponding ground-truth.

| Model | #Param. (M) | GFLOPS | Metrics | | |
| --- | --- | --- | --- | --- | --- |
| | | | $C$-Nino3.4-M $\uparrow$ | $C$-Nino3.4-WM $\uparrow$ | MSE $(10^{-4})\downarrow$ |
| Persistence | - | - | 0.3221 | 0.447 | 4.581 |
| UNet [43] | 12.1 | 0.4 | 0.6926 | 2.102 | 2.868 |
| ConvLSTM [36] | 14.0 | 11.1 | 0.6955 | 2.107 | 2.657 |
| PredRNN [45] | 23.8 | 85.8 | 0.6492 | 1.910 | 3.044 |
| PhyDNet [13] | 3.1 | 5.7 | 0.6646 | 1.965 | 2.708 |
| E3D-LSTM [44] | 12.9 | 99.8 | 0.7040 | 2.125 | 3.095 |
| Rainformer [3] | 19.2 | 1.3 | 0.7106 | 2.153 | 3.043 |
| Earthformer w/o global | 6.6 | 23.6 | 0.7239 | 2.214 | 2.550 |
| Earthformer | 7.6 | 23.9 | **0.7329** | **2.259** | **2.546** |

## 4.3 ICAR-ENSO Sea Surface Temperature Anomalies Forecasting

El Niño/Southern Oscillation (ENSO) has a wide range of associations with regional climate extremes and ecosystem impacts. ENSO sea surface temperature (SST) anomalies forecasting for lead times up to one year (12 steps) is a valuable and challenging problem. Nino3.4 index, which is the area-averaged SST anomalies across a certain area (170°-120°W, 5°S-5°N) of the Pacific, serves as a crucial indicator of this climate event. The forecast quality is evaluated by the correlation skill [15] of the three-month-moving-averaged Nino3.4 index $C^{\text{Nino3.4}} = \frac{\sum_N (\boldsymbol{X} - \bar{\boldsymbol{X}})(\boldsymbol{Y} - \bar{\boldsymbol{Y}})}{\sqrt{\sum_N (\boldsymbol{X} - \bar{\boldsymbol{X}})^2 \sum_N (\boldsymbol{Y} - \bar{\boldsymbol{Y}})^2}} \in \mathbb{R}^K$ calculated on the whole test set of size $N$, where $\boldsymbol{Y} \in \mathbb{R}^{N \times K}$ is the ground-truth of $K$-step Nino3.4 index, $\boldsymbol{X} \in \mathbb{R}^{N \times K}$ is the corresponding prediction of Nino3.4 index.

ICAR-ENSO consists of historical climate observation and stimulation data provided by Institute for Climate and Application Research (ICAR). We forecast the SST anomalies up to 14 steps (2 steps more than one year for calculating three-month-moving-average) given a context of 12 steps of SST anomalies observations. Table 7 compares the performance of our Earthformer with baselines on the ICAR-ENSO dataset. We report the mean correlation skill $C$-Nino3.4-M $= \frac{1}{K} \sum_k C_k^{\text{Nino3.4}}$ and the weighted mean correlation skill $C$-Nino3.4-WM $= \frac{1}{K} \sum_k a_k \cdot C_k^{\text{Nino3.4}}$ over $K = 12$ forecasting steps[5], as well as the MSE between the spatiotemporal SST anomalies prediction and the corresponding ground-truth. We can find that Earthformer consistently outperforms the baselines in all concerned evaluation metrics and that using global vectors further improves the performance.

## 5 Conclusions and Broader Impact

In this paper, we propose Earthformer, a space-time Transformer for Earth system forecasting. Earthformer is based on a generic and efficient building block called *Cuboid Attention*. It achieves SOTA on MovingMNIST, our newly proposed $N$-body MNIST, SEVIR, and ICAR-ENSO.

Our work has certain limitations. The first one is that Earthformer is a deterministic model that does not model uncertainty. This may result in predicting the average of all plausible futures, causing the model to generate blurry predictions of low perceptual quality and be lack of valuable small-scale details. In fact, the community lacks appropriate metrics that measure the uncertainty component in Earth system forecasting models. Extending Earthformer to a probabilistic forecasting model can be an exciting future direction. We include more detailed discussions and preliminary experiments about handling uncertainty in Appendix C. The second one is that the model is purely data-driven and does not take advantage of the physical knowledge of the Earth system. Recent studies on adding physical constraints [20, 26] and ensembling the predictions from a data-driven model and a physics-based model [32] imply that it is an active and promising research direction to pursue. We plan to study how to incorporate physical knowledge into Earthformer in the future.

## Acknowledgments and Disclosure of Funding

This work has been made possible by a Research Impact Fund project (R6003-21) and an Innovation and Technology Fund project (ITS/004/21FP) funded by the Hong Kong Government.

---

[5]$a_k = b_k \cdot \ln k$, where $b_k = 1.5$, for $k \leq 4$; $b_k = 2$, for $4 < k \leq 11$; $b_k = 3$, for $k > 11$.

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
