# OpenReview forum: "Earthformer: Exploring Space-Time Transformers for Earth System Forecasting"
_NeurIPS.cc/2022/Conference — NeurIPS 2022 Accept_

### Official Review · Reviewer_YCyZ · 2022-07-09

**Rating:** 6
**Confidence:** 4
**Soundness:** 3 good
**Presentation:** 3 good
**Contribution:** 3 good

**Summary:**

In this paper, the authors propose an earth system forecast model based on transformers. This is achieved by first dividing the entire space-time field as non-overlapping cuboid blocks, applying attention in each cuboid block together with several "global vectors" that allow for interactions between the individual blocks, and then merging the blocks together to yield the predictions. They find state-of-the-art results on several synthetic and real datasets including moving MNIST, N-body MNIST (an extension of moving MNIST), precipitation nowcasting and SST anomaly forecasting.

**Questions:**

There are several lines throughout the text that I found difficult to comprehend. Please could you clarify this?
- Lines 238-240:
> Note that the MovingMNIST performance on several papers [13] are obtained by training the model with on-the-fly generated digits while we pre-generate the digits and train all models on a fixed dataset.

What are "on-the-fly generated digits"? What was the reasoning behind training the model differently in this paper vs the other papers?

- Lines 184-185:
> generates the prediction via a coarse-to-fine procedure

What exactly is the coarse-to-fine procedure?

- What do the subscripts $(a,b,c)$ in the latent visual code $Z_{(a,b,c)}^{\text{code}}$ mean in the VQ-VAE? This notation is not introduced. Also, shouldn't $Z^{\text{code}}$ in the LHS of equation (9), Appendix D be $Z_i^{\text{code}}$ (since the RHS has index $i$)?

- Could you please explain in more details the idea behind auto-regressive Earthformer? This is to incorporate temporal correlations? But if so, shouldn't the global vectors already take that into account?




**Limitations:**

One limitation is that the model may have good performance with respect to standard machine learning metrics, however in practice, this may not necessarily translate to their usability, as demonstrated in the recent Deepmind paper on precipitation nowcasting (Ravuri et al. 2021). This is already visible in Figures 7-10, where the blurry predictions have better MSE score than the more realistic, high fidelity predictions that are clearly better. This indicates a search for better metrics for earth system forecasting or approaches that don't rely on loss optimisation.

**Strengths And Weaknesses:**

Strengths:
- Thorough benchmarking to demonstrate the performance on real and synthetic data in addition to nice ablation studies.
- Results on the N-body MNIST look great. Predictions are much less blurry than the baselines, which are a common problem for these data-driven nowcasting/forecasting models.

Weaknesses:
- In terms of novelty, with so many recent papers applying transformers to vision tasks, the idea itself is not very original. There are many papers that applies transformers to video prediction tasks, e.g. "Anticipative Video Transformer (Girdhar and Grauman 2021)", that the authors also acknowledge. However, I have not seen any paper that applies cuboid attention, so this paper may be novel in this respect.
- Comparisons with physics-based models would have been nice to see as for many problems in weather forecasting, physics-based model still perform better than data-driven models. It would be interesting to know how the transformer approach fare against these traditional modelling approaches.

---

> ### Author Response · Authors · 2022-08-02
> **Thanks for the feedback and below are our detailed responses (2/2)**
>
> **Q4: explain in more details the idea behind auto-regressive Earthformer**
>
> As shown in Fig. 6 in the Appendix, Earthformer AR uses exactly the same encoder as Earthformer while uses a different auto-regressive (AR) decoder. The encoder uses the global vectors while the AR decoder does not use the global vectors. As explained in Appendix D, The auto-regressive approach encode the input and target into discrete visual tokens. We pretrain a VQ-VAE to convert the input and target to visual codes. We traverse the visual codes in the target in the raster-scan order, and factorized $p(\mathcal{Z}^{\text{code}} \mid \mathcal{X})$ as $\prod\_{(a,b,c)} p\left(\mathcal{Z}^{\text{code}}\_{(a,b,c)}|\mathcal{Z}^{\text{code}}\_{<(a,b,c)}, \mathcal{X}\right)$. The model is then trained via teacher forcing. In the generation phase, the visual codes are generated one by one following the raster-scan order via either argmax or random sampling. AR is widely used for building Transformer-based generative models. As demonstrated by Figure 7-10, predictions from Earthformer AR are more perceptually satisfying. However, the performance of Earthformer AR is much worse in terms of skill scores. We thus included the results from the non-auto-regressive version of Earthformer in the main paper and leave the research on how to better evaluate Earth system forecasting models in the future work.
>
> **L1: evaluation metrics and perceptual quality**
>
> Thanks for pointing it out. We will add extra discussion on the evaluation metrics and perceptual quality as limitations and future works in the Conclusion section. We have done some preliminary trials on these topics in Appendix D. We agree with the reviewer that
> the results suggest that we need to `"search for better metrics for earth system forecasting or approaches that don't rely on solely on loss optimisation"`.
> The DeepMind paper resorted to meteorologists for human evaluations but the community still need cheaper alternatives. The deep learning community has met similar problems when trying to evaluate GAN-based image generative models. Computer vision researchers now rely on scores like Inception Score (IS) and Fréchet Inception Distance (FID) for evaluation. The challenge is that both scores depend on pretrained backbones on ImageNet which **does not exist** for Earth science datasets.
>
> We feel that one solution is to pretrain an auto-encoder on Earth observation data, extract the features with the auto-encoder, and calculate FID. This will be a type of domain-specific FID, which exists in other domains such as [Fréchet Audio Distance](https://arxiv.org/pdf/1812.08466.pdf), [Fréchet Video Distance](https://arxiv.org/abs/1812.01717), and [Fréchet ChemNet Distance](https://pubs.acs.org/doi/10.1021/acs.jcim.8b00234). However, this is out of the scope of this submission. In this submission, we stick to the same evaluation metrics suggested by SEVIR and the ENSO forecasting paper. We decide to explore how to better evaluate the models in the future.

---

> ### Author Response · Authors · 2022-08-02
> **Thanks for the feedback and below are our detailed responses (1/2)**
>
> Thank you for providing detailed feedback. We hope that our response below can address your main concerns.
>
> **W1: in terms of novelty**
>
> We have done literature review on related works from three perspectives: deep learning architectures for Earth system forecasting, space-time Transformers for video modeling, and global and local attention in vision Transformers (line 84-117).
> To the best of our knowledge, we are the first to propose a space-time Transformer that combines structure-aware local attention mechanism and global vectors for Earth system forecasting.
> The paper ["Anticipative video transformer. CVPR 2021"](https://arxiv.org/abs/2106.02036) mentioned by the reviewer **is not a work for video prediction** but for future action anticipation. The predictions generated by the model are **action class labels** rather than the future video frames.
>
> **W2: comparison to physics-based (traditional) methods**
>
> Thanks for pointing it out. This is one limitation of this work. However, we are following the peer-reviewed benchmark papers (e.g., SEVIR is a benchmark paper from NeurIPS 2020) in comparing the performance of the models. We feel that performance comparison with physics-based models is a gap in this area, partially because it is difficult to access these models and obtain the required computational resources to run these physics-based models. In areas where deep learning models are useful, e.g., precipitation nowcasting, it is even difficult to obtain high-resolution predictions from the physical models. We will clarify the limitation in the revision.
>
> **Q1: what are "on-the-fly generated digits"?**
>
> In our MovingMNIST experiments, we pre-generate the training set that contains 8,100 sequences. In the released code of some baseline models (e.g., [the source code of PhyDNet](https://github.com/vincent-leguen/PhyDNet/blob/23a992d771c9eb1d32f52b1873a3c5625f1a8413/data/moving_mnist.py#L118-L121)), sequences are generated on-the-fly and the model is essentially trained with an **unlimited** number of MovingMNIST sequences.
> By training on on-the-fly generated sequences, the model can see way more training sequences than 8,100, and may potentially leak test data. Therefore, directly using the scores reported in the original papers leads to unfair comparison, which is why we chose to rerun all the experiments. Training on a limited number of sequences is also closer to the real-world setting. For instance, SEVIR contains a fixed collection of training sequences from 2017-2019.
>
> **Q2: what exactly is the coarse-to-fine procedure?**
>
> In Sec. 3.3, we propose to use a hierarchical encoder-decoder architecture, where the spatial resolutions of the latent states gradually decrease in the encoder and **gradually increase** in the decoder. "Coarse-to-fine" refers to the resolution changes in the decoder. Table 11 in Appendix B demonstrates the superiority of such hierarchical architectures over non-hierarchical architectures. We observed that the hierarchical architectures have similar FLOPS to non-hierarchical architectures while performing better in terms of MSE.
>
> **Q3.1: what do the subscripts in the latent visual code $\mathcal{Z}^{\text{code}}\_{(a,b,c)}$ mean in the VQ-VAE?**
>
> The subscripts $(a,b,c)$ are the spatiotemporal indices of the discrete encodings. For example, suppose we encode a $10\times 64 \times 64$ MovingMNIST sequence via a VQ-VAE encoder into a $10 \times 16 \times 16$ tensor $\mathcal{Z}^{\text{code}}$ containing latent codes. Each latent code represents the discrete code for a $4 \times 4$ patch, and we can traverse the discrete codes in $\mathcal{Z}^{\text{code}}$ as $\mathcal{Z}^{\text{code}}\_{(0,0,0)}, \mathcal{Z}^{\text{code}}\_{(0,0,1)}, \dots, \mathcal{Z}^{\text{code}}\_{(9,15,15)}$. Here, we refer $\mathcal{Z}^{\text{code}}\_i$ as the discrete codes for the $i$-th frame. The VQ-VAE is trained on 2D images.
>
> **Q3.2: equation (9) in Appendix D**
>
> Sorry for the typo in Eqn. 9 in Appendix D. The LHS should be revised from $p(\mathcal{Z}^{\text{code}})$ to $p(\mathcal{Z}^{\text{code}} \mid \mathcal{X})$. We factorize the joint probability distribution $p(\mathcal{Z}^{\text{code}} \mid \mathcal{X})$ as the product of conditional distributions, i.e., $p\left(\mathcal{Z}^{\text{code}}\_{(a,b,c)} \mid \mathcal{Z}^{\text{code}}\_{<(a,b,c)}, \mathcal{X}\right)$s. We traverse the discrete codes with the raster-scan order so $\mathcal{Z}^{\text{code}}\_{(a,b,c)}$ should condition on all the previously generated codes $[\mathcal{Z}^{\text{code}}\_{<(a,b,c)}]$ and the whole input sequence $\mathcal{X}$.
> The correct Eqn. 9 should be $p(\mathcal{Z}^{\text{code}} \mid \mathcal{X})=\prod\_{(a,b,c)}p\left(\mathcal{Z}^{\text{code}}\_{(a,b,c)} \mid \mathcal{Z}^{\text{code}}\_{<(a,b,c)}, \mathcal{X}\right)$. We will revise Appendix D to fix the typo in the final version.

---

### Official Review · Reviewer_fTQV · 2022-07-10

**Rating:** 6
**Confidence:** 4
**Soundness:** 2 fair
**Presentation:** 3 good
**Contribution:** 3 good

**Summary:**

The authors propose a spacetime transformer and apply it to forecasting events in Earth systems. In detail:
- the authors propose Cuboid attention mechanism which splits the input tensor to non-overlapping parts
  - to reduce computational cost by local attention
  - to compute self-attention in parallel
- the authors generalize previous studied video transformer by stacking cuboid attention layers with various hyperparameters
- the authors fulfill the communication gaps of local Cuboid by introducing shared global vectors
- the authors propose an N-body MNIST dataset developed from MINIST
- the proposed Earthformer achieves SOTA performance on both SEVIR and ICAR-ENSO

**Questions:**

- I am not sure why N-body MNIST is related to the chaotic 3-body motion problem and thus performing good on this dataset means the model is capable to handle complex system. It would be better if the authors illustrate the reasons and intuitions behind this?
- It would be better if you can give some training details. For example, how you encode the input data and what is the dimension of the embeddings.


**Limitations:**

- The authors do not address the limitation adequately
- This framework is not physical dynamic-aware such as the PhyDNet paper which is also mentioned by the authors. To handle a complex system, I think it is important to introduce knowledge from the dynamics.
- There are few analyses showing why this framework shall only be specialized on this task. They might extend this framework to video frame prediction tasks and comparing to the SOTA in these datasets (e.g. UCF101).
- It would be nicer if they can post some failure cases and discuss the possible issues.

**Strengths And Weaknesses:**

Strengths:
 - originality: From my knowledge, it is one of the first paper working on predicting ICAR-ENSO using transformer.
 - quality: The authors use both synthetic dataset--N-body MNIST and SEVIR, ICAR-ENSO to proof the significance improvement of their work.
 - clarity: The Cuboid Attention is introduced very clearly with formula and detailed figure illustration
 - significance: The improvement of the performance on the model is significant

Weaknesses:
 - originality: There are a lot of work of transformers with local attention and shared global connection between these local attentions. The model architecture is not novel from a machine learning model aspects.
 - quality: It would be nicer if the author can use other transformer based space-time model to train on these benchmarks and compare the performance with them. After all, ConvLSTM is from 2015, PredRNN and PhyDNetis not based on transformer, so does E3D-LSTM which is from 2017. Rainformer is the only one using transformer, but not specialized in this task.

---

> ### Author Response · Authors · 2022-08-02
> **Thanks for the feedback and below are our detailed responses (2/2)**
>
> **Q1: N-body and complex system**
>
> Thanks for the question. Indeed, good performance on N-body MNIST does not guarantee that the model can handle complex systems well. However, failing to perform well on N-body MNIST suggests that the model is not able to correctly learn the long-range and non-linear interactions that are prevalent in Earth systems. Evaluating on a synthetic dataset with sufficiently sophisticated motion **helps us figure out what kinds of dynamics the models fail to handle**. Our N-body MNIST is more challenging than the widely used MovingMNIST dataset as shown in Figure 11 in Appendix E. In contrast, it is much more difficult to conclude from the failures on the real-world datasets on what kind of flaws the models have; there are many factors in real-world datasets: the data can sometimes be noisy, and the system dynamics is usually complicated and even unknown.
>
> **Q2: training details**
>
> Due to space limitation, we have included the implementation details in **Appendix A**. Specifically, Appendix A.1 and Table 7,8,9 provide details about Earthformer. Appendix A.2 and Table 10 provide details about the baselines we compared (UNet, ConvLSTM, PredRNN, PhyDNet, E3D-LSTM, Rainformer). We also included the source code (which contain all training details) in the supplementary materials and will release it once the paper gets accepted.
>
> **L1,L4: limitations and failure cases**
>
> Thanks for pointing it out. We will emphasize the limitations in the "Conclusion" section in the final version of the paper. For example, we will move the brief discussion in **Appendix D** around the limitation of being difficult to train a model that has both good perceptual quality and evaluation metrics to the main text. We will also state that the current model lacks physical knowledge and how we are planning to incorporate physical constraints in the future work.
>
> **L2: physics-based methods and domain knowledge**
>
> Thanks for your suggestion. Incorporating physical knowledge in the neural network and figuring out an appropriate network architecture can be two orthogonal topics. You may view Earthformer as a neural network that is flexible enough for learning the physics from data. Under this framework, there are two ways to incorporate knowledge from a physical system: 1) Along with the earth observations, take a physical simulator's output as the input of Earthformer. This essentially ensembles the data-driven DL model and a physics-driven model and has shown to perform well in [9]; 2) Incorporate physical knowledge in the neural network as a type of physical constraints, as suggested in "[Physics-informed neural networks](https://www.sciencedirect.com/science/article/pii/S0021999118307125)". This is an active research area and people have shown in a recent NeurIPS publication ["Characterizing possible failure modes in physics-informed neural networks"](https://papers.nips.cc/paper/2021/hash/df438e5206f31600e6ae4af72f2725f1-Abstract.html) the limitations of the traditional soft constraint regularizing approach, and enforcing the physics as hard constraints has been recently studied in ["Learning differentiable solvers for systems with hard constraints"](https://arxiv.org/pdf/2207.08675.pdf). Based on the reasoning, we will leave the investigation on incorporating physical knowledge as the future work.
>
>
> **L3: extend to video prediction**
>
> *Note: this is similar to "W1: missing some experimental datasets" under Reviewer LEpa. You may also check our response there.
>
> Thanks for your suggestion. We will leave the comparison on some video frame prediction datasets in the future work. For this paper, we focus on evaluating the model on Earth system forecasting tasks (precipitation nowcasting and ENSO forecasting) because they have concrete social impact and real-world application scenarios. We feel that for action recognition datasets like UCF101, the general goal is to generate photo-realistic videos. This can be different from the goal of Earth system forecasting, which is to generate accurate predictions. Photo-realistic generation of videos may require a different set of methods like GAN / denoising diffusion model, and metrics like Inception Score (IS) or Fréchet Inception Distance (FID). We plan to extend Earthformer with GAN or denoising diffusion model (mentioned in the Conclusion section of the submission) and revisit video frame prediction in the follow-up work. For this paper, we stick to the problem of exploring space-time Transformers for Earth system forecasting, which has not been studied yet.

---

> ### Author Response · Authors · 2022-08-02
> **Thanks for the feedback and below are our detailed responses (1/2)**
>
> Thank you for providing detailed feedback. We hope that our response below can address your main concerns.
>
> **W1: ...lot of work of transformers with local attention and shared global connection between these local attentions. The model architecture is not novel...**
>
> Indeed there are papers introducing global vectors to help information exchange in local attention and we have reviewed them in Sec. 2 "Global and local attention in vision Transformers. " (Line 107-117) and stated our novelty. To the best of our knowledge, we are the first to point out the importance of adding global vectors in space-time Transformers for spatiotemporal forecasting. In addition, our cuboid attention is new and can be viewed as providing a ``"common language for discussing various forms of attention"`` as suggested by Reviewer LEpa. As pointed out by Reviewer fTQV, by introducing shared global vectors, we ``"fulfill the communication gaps of local cuboids"``. From the results in Table 3-6, we can see that global vectors are crucial for successful spatiotemporal forecasting and bring consistent performance improvement for a wide range of space-time attention patterns on different problems. To the best of our knowledge, we haven't seen other papers adopt the same generic "cuboid + global vector" formulation and we are novel in this respect.
>
> **W2: Transformer-based baselines**
>
> Though there have been a number of works in recent years that propose space-time Transformers for video understanding, only a limited number of them have studied the spatiotemporal forecasting problem, in which both the input and output are videos. As an alternative, we adopt cuboid attention to subsume three previously proposed efficient space-time Transformers ("Axial", "Divided Space-Time" and "Video Swin") into our encoder-decoder architecture. Comparison among different space-time attentions on MovingMNIST and N-body MNIST are summarized in **Table 3**. We also picked Rainformer as a baseline since it is a recently proposed Transformer-based method for precipitation nowcasting. Earthformer outperforms Rainformer according to Table 5 and 6.
>
> In addition, we discussed the non-auto-regressive (NAR) and auto-regressive (AR) variants of Earthformer in Appendix D. The auto-regressive variant of Earthformer is inspired by ["VideoGPT: Video generation using VQ-VAE and Transformers"](https://arxiv.org/abs/2104.10157) and ["Latent Video Transformer"](https://arxiv.org/abs/2006.10704). Different from these papers, we use cuboid attention as the building block and adopt a hierarchical decoder. The outputs of Earthformer-AR look more like "real" SEVIR images and do not suffer from blurry-prediction issues. However, the performance of Earthformer-AR is much worse than Earthformer in terms of the skill scores (even worse than baselines like U-Net). We thus included the results from the NAR variant in the main paper. The discrepancy between perceptual quality and skill scores is an open problem in this area and certainly requires future work.
>
> Like what has been pointed out by Reviewer YCyZ, the results suggest that we need to `"search for better metrics for earth system forecasting or approaches that don't rely on solely on loss optimization"`. The DeepMind paper [30] resorted to meteorologists for human evaluations but the community still need cheaper alternatives. The deep learning community has met similar problems when trying to evaluate GAN-based image generative models. Computer vision researchers now rely on scores like Inception Score (IS) and Fréchet Inception Distance (FID) for evaluation. The challenge is that both scores depend on pretrained backbones on ImageNet which **does not exist** for Earth science datasets. We feel that one solution is to pretrain an auto-encoder on Earth observation data, extract the features with the auto-encoder, and calculate FID. This will be a type of domain-specific FID, which exists in other domains such as [Fréchet Audio Distance](https://arxiv.org/pdf/1812.08466.pdf), [Fréchet Video Distance](https://arxiv.org/abs/1812.01717), and [Fréchet ChemNet Distance](https://pubs.acs.org/doi/10.1021/acs.jcim.8b00234). However, this is out of the scope of this submission. In this submission, we stick to the same evaluation metrics suggested by SEVIR and the ENSO forecasting paper. We decide to explore how to better evaluate the models in the future.

---

> ### Comment · Reviewer_fTQV · 2022-08-10
> **Response to Paper4429 Authors**
>
> Thank you for your response.
>
> W1: The reviewer still holds the viewpoint that similar ideas have been pointed out (e.g. Coordination Among Neural Modules Through a Shared Global Workspace: https://arxiv.org/abs/2103.01197). Based on that idea, even if the architecture implemented is novel, it is still more like an engineering details problem.
>
> W2: The authors mentioned "The challenge is that both scores depend on pretrained backbones on ImageNet which does not exist for Earth science datasets." I think the reviewer could ask for some human expert evaluations or if the task is not that hard, you could use some platforms such as Amazon Mechanical Turk to have a good evaluation on your model output. Even if this is not compulsory, this should give reviewers more confidence on your model's results.
>
> Thanks for answering Q1, Q2. The reviewer decided to keep the rating unchanged. Thank you!

---

### Official Review · Reviewer_LEpa · 2022-07-11

**Rating:** 6
**Confidence:** 3
**Soundness:** 3 good
**Presentation:** 3 good
**Contribution:** 3 good

**Summary:**

This paper proposes the framework of "cuboid attention" for thinking about different spatio-temporal attention mechanisms. This framework includes vanilla self-attention and various lower-complexity alternatives such as axial attention or divided space-time attention as special cases.

Furthermore, this paper proposes Earthformer, a spatio-temporal transformer model with a particular set of cuboid attention blocks. The Earthformer architecture further features a set of global vectors that all cuboids attend to in order to more-readily share information between cuboids.

This paper also proposes a dataset called N-body MNIST (with N=3), where each dataset example is a sequence 64x64 frames consisting of 3 MNIST digits moving according to chaotic 3-body motion. This dataset contains 22,000 examples.

Finally, the authors evaluate the Earthformer architecture against several existing algorithms (UNet, ConvLSTM, PredRNN, PhyDNet, E3D-LSTM, and Rainformer) on several datasets (MovingMNIST, N-body MNIST, SEVIR, and ICAR-ENSO). The results show that Earthformer consistently outperforms the other models, sometimes by a large margin.

**Questions:**

**Q1**: To me, it was unclear whether the results of baseline models reported in Tables 4-6 were computed by the Earthformer authors, or the original authors of those baseline models. Could the Earthformer authors please clarify?

**Q2**: Will the N-body MNIST dataset be released publicly?

**Q3**: Why call the dataset "N-body MNIST" instead of "3-body MNIST" if you only tried N=3? Given that you only tried N=3, I think it's a bit misleading to write N-body.

**Q4**: You write for cuboid attention on its own, "the cuboids do not communicate with each other." This sentence is a bit vague, and perhaps misleading. If divided space-time attention is applied sequentially (first space, then time), then there is information passing between all cuboids, right?

**Limitations:**

To me, there are no clear limitations / negative impact. The authors did not discuss any negative impact.

**Strengths And Weaknesses:**

## Strengths

**S1: New N-body MNIST dataset**\
Compared to MovingMNIST, the N-body MNIST dataset intuitively seems more challenging and representative of realistic dynamical systems. I believe that this dataset could well serve as a standard benchmark for future models.

**S2: Comprehensive ablation study on the benefit of global attention vectors**\
I appreciated the large amount of ablation studies showing the significant performance gain with using global attention vectors on the MovingMNIST and N-body MNIST datasets, regardless of the attention variant.

**S3: Generalization of attention via Cuboid Attention framework**\
The cuboid attention framework for thinking about attention mechanisms serves to provide a common language for discussing various forms of attention. In particular, the authors show how axial attention and divided space-time attention are special cases of cuboid attention.

**S4: Overall impressive results**\
The comparison tables and figure really demonstrate the superiority of Earthformer against the baseline models.

## Weaknesses

**W1: Missing some experimental datasets**\
While the authors show good results of Earthformer on MovingMNIST, N-body MNIST, SEVIR, and ICAR-ENSO, I would be interested in seeing experiments on other types of spatio-temporal datasets, such as action-recognition (Kinetics-400/600, SSV2, Diving-48, or YouTube-8M).

**W2: Missing some comparison models**\
While the set of model comparisons is large, to me it seems that some more generic models are missing. For example, how would a standard TimeSformer model (with divided space-time attention) do on the experiments? A comparison against TimeSformer, for example, would really help cement the importance of (1) cuboid attention, and (2) global attention vectors. An appropriate comparison would also be against TimeSformer, but with global attention added to the TimeSformer model.

---

> ### Author Response · Authors · 2022-08-02
> **Thanks for the feedback and below are our detailed responses (2/2)**
>
> **Q1: It was unclear whether the results of baseline models reported in Tables 4-6 were computed by the Earthformer authors, or the original authors of those baseline models**
>
> For synthetic datasets, as stated in Sec. 4.1 (line 238-242), we **train all baselines from scratch** using the default hyperparameters and configurations in their officially released code for fair comparison. As mentioned in the paper (line 238-242), the reason for rerunning the experiments instead of copying the results is that several papers train the models with on-the-fly generated digits (e.g., [the source code of PhyDNet](https://github.com/vincent-leguen/PhyDNet/blob/23a992d771c9eb1d32f52b1873a3c5625f1a8413/data/moving_mnist.py#L118-L121)), while we pre-generate the digits and train all models on a fixed dataset. Training on a fixed dataset is closer to the real-world scenario. For instance, SEVIR contains a fixed collection of training sequences from 2017-2019. We plan to release all codes for reproducing the baselines and our Earthformer model once the paper gets accepted.
>
> For SEVIR and ICAR-ENSO, as stated in Appendix A.2, we follow the officially released configurations of the baselines and **tune the hyperparameters** to optimize their performance on each dataset. Detailed configurations are shown in Table 10 of the Appendix.
>
> **Q2 + Q3: Will the N-body MNIST dataset be released publicly? "N-body MNIST" instead of "3-body MNIST"**
>
> The code for generating the N-body dataset will be released publicly once the paper get accepted. You can find the code in our supplementary materials (`"generate_nbody_dataset.py"` and `"src/earthformer/nbody_mnist.py"`).
>
> We name the dataset "N-body" to remind the readers of the chaotic behavior in N-body systems. Our implementation also supports $N>3$. In our preliminary experiments, we find that $N=3$ is already challenging enough that all baselines fail to give precise predictions.
>
>
> **Q4: communications among cuboids**
>
> Thanks for pointing out the ambiguity. By "the cuboids do not communicate with each other", we mean that inter-cuboid communication does not happen in a **single layer** without global vectors. When stacking multiple cuboid attention layers with different decomposition patterns, the later layer can have a more global view of the system. Nevertheless, adding extra global vectors facilitates information exchange among cuboids within each layer. This provides each cuboid layer with stronger representational power. With these global vectors, global information can be captured in each layer, **even without stacking layers with different cuboid patterns**. According to our experiments, this consistently boosts the performance.

---

> ### Author Response · Authors · 2022-08-02
> **Thanks for the feedback and below are our detailed responses (1/2)**
>
> Thank you for providing detailed feedback. We hope that our response below can address your main concerns.
>
> **W1: missing some experimental datasets**
>
> Really appreciate your suggestion and we will apply cuboid attention on action-recognition datasets in the future work. From our point of view, although the modeling techniques are similar, video generation for action recognition and Earth system forecasting are two different application scenarios. The former requires the model to generate photo-realistic videos and the latter requires the model to give predictions that have high forecasting skill scores. Obtaining photo-realistic generative model for action-recognition datasets often requires 1) pretraining the backbone on ImageNet and 2) further train the model with GAN loss or use denoising diffusion model. The evaluation method for these two applications are also different. For example, we can evaluate the model's ability of giving photo-realistic generations by checking the Inception Scores (IS) or Fréchet Inception Distance (FID), while Earth system forecasting models are usually evaluated via skill scores like CSI / correlation skills. Since the focus of our submission is on exploring the **architectural design** of space-time Transformers for Earth system forecasting, we feel that additional experiments on action recognition datasets may make the paper less focused.
>
> Thus, we choose to conduct empirical experiments on synthetic datasets (MovingMNIST, N-body MNIST) and two Earth system forecasting tasks. The two Earth system forecasting tasks we picked have concrete real-world application scenarios and benchmark datasets. For example, SEVIR is published in NeurIPS 2020 and ICAR-ENSO is from Nature 2019. Both have been peer-reviewed. In fact, SEVIR is accepted in NeurIPS 2020 because it is a high-quality benchmark. In the [meta-review of the SEVIR paper](https://papers.nips.cc/paper/2020/file/fa78a16157fed00d7a80515818432169-MetaReview.html), the meta-reviewer called out that "a large, high-quality data set for real tasks in atmospheric/earth sciences is important and could spur AI work in this area". We agree with this meta-review and evaluated our model on solid benchmarks like SEVIR and ICAR-ENSO. In the submission, we adopt evaluation metrics recommended by these papers, i.e., CSI for SEVIR, and Nino3.4 index correlation scores for ICAR-ENSO. On the other hand, the synthetic datasets help us interpret the behavior of the model and figure out model flaws. For example, by visualizing the prediction results on the N-body MNIST dataset (Fig. 5), it is easy to notice that Earthformer can outperform other baselines in capturing complicated motion patterns. MovingMNIST has been widely adopted in this domain for model development and evaluation. The N-body MNIST is a follow-up along this effort and is more challenging due to the chaotic 3-body motion.
>
> **W2: Missing some comparison models**
>
> Thanks for the question. TimeSformer is designed for action recognition (the input is a video and the output is a label) and is not directly applicable to spatiotemporal forecasting (both the input and the output are videos). We will leave the extension of cuboid attention on action recognition as future work. Like what the reviewer has pointed out, the proposed cuboid attention ``"provide a common language for discussing various forms of attention"``. Besides the baseline methods listed in Sec. 4.1 (line 237), we compared different variants of cuboid attention patterns shown in **Table 3**. Among them, the "DST" pattern is directly inspired by TimeSformer. Via our framework, we essentially extended TimeSformer to video prediction. In addition, the results in **the 2nd row of Table 3** suggest that our newly proposed global vector improves the performance of the DST pattern. To the best of our knowledge, we are the first to add global vectors to DST. Likewise, the "Video Swin" pattern is inspired by Video Swin Transformer. We have extended Video Swin Transformer to spatiotemporal forecasting and also showed that adding global vectors can boost the performance with minimal additional computation.

---

### Official Review · Reviewer_DF9s · 2022-07-31

**Rating:** 3
**Confidence:** 5
**Soundness:** 2 fair
**Presentation:** 2 fair
**Contribution:** 2 fair

**Summary:**

This paper proposed cuboid attention as an alternative to regular attention for forecasting Earth system-like data. The paper presents results on a toy N-body MNIST dataset and two real datasets, one on precipitation nowcasting and one one El-Nino Southern Oscillation





**Questions:**

1. The motivation behind not using CNN/RNN is that the inductive bias might not capture the internal variability of the climate system. Still, the proposed architecture also has not been studied in terms of why it should.

2. It's important to understand that for problems in physics, a single number like MSE and MAE may not be a good representative of actually how well the algorithm performs in terms of forecasting. For example, in the motivation, the authors talk about how internal climate variability can affect the prediction performance-- that's a solid motivation but is completely absent in the analysis. Does their model capture large scales better or do they capture small scales for a longer period of time? All of these can be very well explored with a Lorenz-type system instead of this MNIST data set. Weather is more than just chaotic--it is multi-scale without any scale separation, unlike the MNIST data

2. The El Nino and nowcasting predictions results need further elaboration. It's hard to say whether it's good or bad. Also, there are good baselines for both these problems. To my knowledge, this paper is SOTA for DL-based ENSO:https://www.nature.com/articles/s41586-019-1559-7. Can the authors compare their model with the proposed model?

3. For precipitation nowcasting, I believe this paper is SOTA:https://www.nature.com/articles/s41586-021-03854-z. Is it possible to compare the results directly against this paper?

4. I would finally also like to point out, that even if the proposed model is not outperforming some of the baselines it does not necessarily take anything away from the architecture, but the authors need to do some analysis and show the usefulness or show the connection between how this architecture is really useful for physical/earth-system data. In the current form, it's not really there

**Strengths And Weaknesses:**

Strength:

The novelty in this paper lies in the fact that they use cuboid attention in their transformer architecture. The motivation is that internal variability in climate may not be captured by the inductive biases of CNN/RNN type architectures. However, there is no reason to think that transformer architectures would also capture internal variability just as much as CNN/RNN or for that matter any DL architecture

Weakness:

This paper has several weaknesses, starting from the motivation behind using cuboid attention and baselines. Furthermore, while I found the N-body system's example as a nice toy example to prove that the proposed algorithm has superiority over other baselines, it is actually not that relevant to weather. I detail it further in my comments.

---

> ### Author Response · Authors · 2022-08-02
> **Thanks for the feedback and below are our detailed responses (4/4)**
>
>
> [1] Ham, Yoo-Geun, Jeong-Hwan Kim, and Jing-Jia Luo. "Deep learning for multi-year ENSO forecasts." Nature 573.7775 (2019): 568-572.
>
> [2] Ravuri, Suman, et al. "Skilful precipitation nowcasting using deep generative models of radar." Nature 597.7878 (2021): 672-677.
>
> [3] Veillette, Mark, Siddharth Samsi, and Chris Mattioli. "Sevir: A storm event imagery dataset for deep learning applications in radar and satellite meteorology." Advances in Neural Information Processing Systems 33 (2020): 22009-22019.
>
> [4] Kilgour, Kevin, et al. "Fr\'echet Audio Distance: A Metric for Evaluating Music Enhancement Algorithms." arXiv preprint arXiv:1812.08466 (2018).
>
> [5] Unterthiner, Thomas, et al. "Towards accurate generative models of video: A new metric & challenges." arXiv preprint arXiv:1812.01717 (2018).
>
> [6] Preuer, Kristina, et al. "Fréchet ChemNet distance: a metric for generative models for molecules in drug discovery." Journal of chemical information and modeling 58.9 (2018): 1736-1741.
>
> [7] Lorenz, Edward N. "Predictability: A problem partly solved." Proc. Seminar on predictability. Vol. 1. No. 1. 1996.

---

> ### Author Response · Authors · 2022-08-02
> **Thanks for the feedback and below are our detailed responses (3/4)**
>
> **Q3: Deep learning for multi-year ENSO forecasts**
>
> We have already cited the reference paper [1] (reference [15] in our paper) mentioned by the reviewer about ENSO forecasting.
> As introduced in the abstract and Fig. 1 on page 1 of [1], the model **is based on CNN** and is designed for Nino3.4 index prediction only.
> In our formulation, we explore the more generic spatiotemporal forecasting setting. Specifically, we predict the future sea surface temperature (SST) anomalies. The Nino3.4 index (line 275), which is the area-averaged SST anomalies, can be directly derived by the predicted SST anomalies. We also report the correlation skill score of the Nino3.4 index. Experiment results in Table 6 demonstrate the advantages of Earthformer over CNN-based models (UNet) in all concerned evaluation metrics.
>
> **Q4: Skillful precipitation nowcasting using deep generative models of radar**
>
> We have already cited the reference paper [2] (reference 28 in our paper) mentioned by the reviewer.
> As shown in "Extended Data Fig. 1" of the paper, the design of DGMR generator proposed in [2] is an encoder-decoder architecture with ConvGRU blocks, which is very similar to ConvLSTM that we have already included as a baseline. DGMR is trained via the GAN loss and has extra discriminators for improving the perceptual quality of the predictions.
> DGMR does give more "realistic" predictions. However, **DGMR does not outperform baseline UNet** in terms of CSI as shown in Fig. 2a of [2]. It is hard to evaluate the perceptual quality and fidelity of the predictions by commonly used metrics. [2] resorts to human evaluation due to the lack of studies on the evaluation metrics in Earth system forecasting literature. In this paper, we choose to stick to evaluating the model with skill scores, which is adopted by SEVIR that is peer-reviewed and published in NeurIPS 2020.
>
> Similar to [2], we also observed that generating more "realistic" predictions does not necessarily mean having higher skill scores. The reviewer may refer to Appendix D for more discussion. In Appendix D, we compared two variants of Earthformer: one with the non-auto-regressive decoder and another with auto-regressive decoder. The auto-regressive variant of Earthformer is inspired by ["VideoGPT: Video generation using VQ-VAE and Transformers"](https://arxiv.org/abs/2104.10157) and ["Latent Video Transformer"](https://arxiv.org/abs/2006.10704). Different from these papers, we use cuboid attention as the building block and adopt a hierarchical decoder. The outputs of Earthformer-AR look more like "real" SEVIR images and do not suffer from blurry-prediction issues. However, the performance of Earthformer-AR is much worse than Earthformer in terms of the skill scores (even worse than baselines like U-Net). We thus focus on exploring the NAR variant in the main paper. The discrepancy between perceptual quality and skill scores is an open problem in this area and certainly requires future works.
>
> For the evaluation problem, one potential solution is to pretrain an auto-encoder on Earth observation data and rely on the combination of Fréchet Inception Distance (FID). There are variants of FID scores in other domains like audio generation [4], video generation [5], and molecule generation [6]. However, this is out of the scope of this paper and should better be future work. In this paper, we are focusing on exploring the space-time Transformers for Earth system forecasting so we stick to evaluating the models with skill scores.
>
> **Q5: show the usefulness**
>
> We do not understand what the reviewer want to point out.
> We have done extensive empirical studies on two synthetic datasets (including a challenging N-body MNIST dataset) to show that Earthformer overcomes concerned drawbacks of existing spatiotemporal forecasting models and are capable of learning complex and long-range dependencies.
> We have done experiments on SEVIR [3], a precipitation forecasting benchmark published at NeurIPS 2020, to demonstrate the effectiveness of Earthformer on precipitation nowcasting task.
> We have also conducted experiment on ENSO forecasting [1] which is also an impactful Earth system forecasting task, to demonstrate the advantages of Earthformer.
> These extensive empirical studies provide strong evidence on `"how this architecture is really useful for physical/earth-system data"`.

---

> ### Author Response · Authors · 2022-08-02
> **Thanks for the feedback and below are our detailed responses (2/4)**
>
> **Q2.4: multi-scale without any scale separation**
>
> We are not sure what the reviewer means by referring to "multi-scale" so we adopt the common definition from computer vision.
> Actually, many real-world dynamic systems are multi-scale and may not have a clear scale separation. Even in video with multiple moving digits, different digits may have different sizes, and the scale is adaptive across the frame. In Earthformer, we used global vectors to connect each local cuboids, and this global-->local attention is not affected by our choice of cuboid decomposition, and is not sensitive to the "scale separation". Also, our hierarchical architecture naturally suits for multi-scale spatiotemporal systems. In Table 11, Appendix B, we demonstrate the superiority of such hierarchical architectures over non-hierarchical architectures. We observed that the hierarchical architectures have similar FLOPS to non-hierarchical architectures while performing better in terms of MSE.
>
> **Q2.5: all of these can be very well explored with a Lorenz-type system instead of this MNIST data set**
>
> We do not understand what the reviewer is suggesting by referring to "Lorenz-type system". Thus, we searched on Google with the keyword "Lorenz-type system,  multi-scale, without any scale separation" and find a paper that seems to be related to this sentence: [**Data-driven predictions of a multiscale Lorenz 96 chaotic system using machine-learning methods: reservoir computing, artificial neural network, and long short-term memory network**](https://npg.copernicus.org/articles/27/373/2020/).
>
> This paper proposed a data-driven model to learn the evolution of $X$ in a Lorenz-type system (a three-tier extension of Lorenz's original model [7]) without the need for knowing the latent factors $Y$ and $Z$. Compared to traditional simulations that require simulating the non-linear evolution of $X,Y,Z$ simultaneously, the data-drive model only needs to know $X$. Also, the paper showed that the Reservoir computing–echo state network (RC–ESN) outperforms artificial neural network (ANN) and LSTM on this Lorenz-96 dataset. We checked the source code of the paper and noticed that the observation and forecasting target $X$ just have **a temporal dimension and a channel dimension** ($C=8$ in the released dataset, See [source code](https://github.com/ashesh6810/RCESN_spatio_temporal/tree/921e60c32b03a60afe111e6ce31a201fe8faff5a).). It is inappropriate to call it "spatiotemporal" from the machine learning perspective where there is only one "channel" dimension with $C=8$. It makes more sense to categorize it as a multivariate time series dataset, which does not belong to the scope of this paper.
>
> In fact, our N-body MNIST dataset has two latent variables: velocity and acceleration that changes following the N-body motion pattern (See the `nbody_mnist.py` file in the supplementary material).
>
> The underlying dynamics in the N-body MNIST dataset is governed by the highly non-linear Newton's law of universal gravitation:
>
> $\frac{d^2\boldsymbol{x}\_{i}}{dt^2} = - \sum\_{j\neq i}\frac{G m\_j (\boldsymbol{x}\_{i}-\boldsymbol{x}\_{j})}{(\|\boldsymbol{x}\_i-\boldsymbol{x}\_j\|+d\_{\text{soft}})^r}$,
>
> where $\boldsymbol{x}\_{i}$ is the spatial coordinates of the $i$-th digit, $G$ is the gravitational constant, $m\_j$ is the mass of the $j$-th digit, $r$ is a constant representing the power scale in the gravitational law, $d\_{\text{soft}}$ is a small softening distance that ensures numerical stability. There are totally $N$ digits interacting with each other in the system. Besides, the digits bounce back when they hit the wall, which makes the dynamics of the dataset even more complex.
>
> We conclude the advantages of conducting empirical studies on N-body MNIST dataset:
> 1. The dynamics in N-body MNIST dataset is intuitive and interpretable.
> 2. N-body MNIST dataset has $N$ systems interacting with each other. The velocity and acceleration of each digits are latent and not observable given the rendered frame. The governing ODE equations ensure that the underlying dynamics is highly non-linear and chaotic.
> 3. N-body MNIST is an extension based on MovingMNIST, which is a widely studied synthetic dataset in video generation. N-body MNIST has exactly the same resolution, input/target sequence lengths, objects appearance, and similar dataset size as MovingMNIST. Hence, it is suitable for using N-body MNIST to study the baselines, which has proved to perform well on MovingMNIST. Failure in the N-body MNIST shows that the previous CNN/RNN-based architectures have limitations.
> 4. N-body MNIST is challenging enough that all studied baselines fail to capture the motion pattern precisely. This sheds light on the issues the previous architectures have.
>
> We thus feel that the N-body MNIST dataset is appropriate for evaluating the model's ability in capturing more complicated motion patterns.

---

> ### Author Response · Authors · 2022-08-02
> **Thanks for the feedback and below are our detailed responses (1/4)**
>
> Thank you for providing feedback. We hope that our response below can address your main concerns.
>
> **Inappropriate summary: the paper presents results on a toy N-body MNIST dataset and two real datasets**
>
> We have to point out the factual errors in the review of reviewer DF9s by summarizing "**a toy** N-body MNIST dataset".
> Reviewer DF9s is suggested to re-read the part of introducing N-body MNIST dataset as it appears that the nature of our proposed N-body MNIST dataset has been misunderstood.
> Reviewer DF9s may also want to read the response to "Q2: all of these can be very well explored with a Lorenz-type system instead of this MNIST data set" for more elaboration of our N-body MNIST dataset.
>
> All other reviewers have summarized our contribution on the **newly proposed** N-body MNIST dataset and experiments on **two synthetic** datasets and two real-world datasets ("This paper also proposes a dataset called N-body MNIST (with N=3)" by reviewer LEpa, "the authors propose an N-body MNIST dataset developed from MINIST" by reviewer fTQV, "They find state-of-the-art results on several synthetic and real datasets including moving MNIST, N-body MNIST (an extension of moving MNIST)" by reviewer YCyZ).
> However, reviewer DF9s **does not distinguish between N-body MNIST and MovingMNIST** and does not acknowledge the complex and chaotic dynamics in N-body MNIST that mimics the dynamics in Earth system (Last two sentences in Q2).
>
> **Q1: motivation behind not using CNN/RNN**
>
> Our motivation for not using CNN/RNN is that the inductive biases may not hold in modeling Earth system (See **line 48**). Transformer is a **more flexible model due to the attention mechanism**. The combination of multi-head attention and feed-forward network makes Transformer good at modeling complex and long-range dependencies. Recent progress in NLP [7] and vision [20] suggests that Transformer is a promising alternative to CNN/RNNs (See **line 49-58**). Although no prior work has systematically studied how Transformer performs for Earth system forecasting, our experimental results (Table 3-6) provide strong evidence on the advantage of Transformer over CNN/RNN for this problem.
>
> **Q2.1: a single number like MSE and MAE may not be a good representative**
>
> Besides MSE/MAE, we also include metrics recommended in SEVIR and ENSO forecasting, such as **CSI** and **correlation skill** (See **Table 5,6**). Actually, SEVIR is peer-reviewed and accepted in NeurIPS 2020. In the [meta-review of the SEVIR paper](https://papers.nips.cc/paper/2020/file/fa78a16157fed00d7a80515818432169-MetaReview.html), the meta-reviewer called out that "a large, high-quality data set for real tasks in atmospheric/earth sciences is important and could spur AI work in this area". We agree with this meta-review and we follow the same method as the paper to evaluate different models.
>
> **Q2.2: how internal climate variability can affect the prediction performance**
>
> We do not understand the reasoning of the reviewer. For example, when training on the sequences in N-body MNIST, the model only sees the displayed pixels and won't know the internal states (i.e., velocity, acceleration) of the moving digits. In order to make accurate predictions, the model has to infer the internal states of the system from the observed pixels. We demonstrate that Earthformer avoids the failure cases of CNN / RNN based models in N-body MNIST in Fig. 5. In addition, experiments show that Earthformer achieves better scores (MSE, MAE, CSI, correlation skill) than baselines in SEVIR and ICAR-ENSO. These serve as strong evidence that Earthformer overcomes the limitations of CNN/RNN that may be due to inductive biases.
>
> **Q2.3: capture large scales better or do they capture small scales for a longer period of time**
>
> We do not understand what the reviewer is referring to here. For example, for SEVIR, we predict for 12 steps ahead (60 minutes) in the future. For the ICAR-ENSO, we predict for 12 steps ahead (1 year) in the future. Details of the datasets are in Table 2. On both datasets, the final skill scores are averaged for all the 12 steps.

---

> ### Author Response · Authors · 2022-08-08
> **Rebuttal discussion**
>
> We hope our responses can resolve your concerns. And we look forward to further questions/concerns (if any), which we would be more than happy to address.

---

### Author Response · Authors · 2022-08-02
**General Response**

Dear reviewers,

We thank all reviewers’ effort in evaluating this work.
We are glad that they found our paper writing is "clear" and "detailed" (fTQV), our cuboid attention "provides a common language for various attention" (LEpa) and "novel" (YCyZ, DF9s), our newly proposed N-body MNIST dataset is "representative" (LEpa) and the results on it "look great" (YCyZ), our empirical studies are "comprehensive" (LEpa), "impressive" (LEpa) and "thorough" (YCyZ), the improvement of the performance is "significant" (fTQV).

In summary, our paper proposes Earthformer, a space-time Transformer based on a novel, generic and flexible Cuboid Attention block, for Earth system forecasting. Cuboid attention provides a common language for various existing space-time attention mechanisms and we introduced the idea of adding global vectors to enhance communication among the cuboids. We propose a new chaotic N-body MNIST dataset extended from MovingMNIST to reveal the limitations of previous methods. We conduct extensive empirical studies to evaluate Earthformer on two synthetic datasets and two real-world datasets about precipitation nowcasting and ENSO forecasting. Earthformer achieves state-of-the-art performance on all datasets and adding global vectors give consistent improvement.

We thank all reviewers' feedback on the limitations of this work. We will modify the "Conclusion" section to "Conclusion and Broader Impact" and discuss the limitations in our final revision. In the following, we list the major limitations / revision of the paper:
1. Exploring how to appropriately evaluate the Earth system forecasting models can be an interesting future direction, but is beyond the scope of this paper. We briefly discussed the topic in Appendix D but will move the discussion in main paper. See details in our reply to Reviewer fTQV "W2: Transformer-based baselines", and Reviewer YCyZ "L1: evaluation metrics and perceptual quality".
2. We will emphasize that the current Earthformer does not incorporate physical constraints and discuss how to extend it to incorporate physical constraints (See details on our reply to Reviewer fTQV "L2: physics-based methods and domain knowledge"). Incorporating physical constraints and exploring the design of neural network architectures can largely be orthogonal topics.
3. There are typos in Appendix D, Eqn.9. We will correct them as in the response to Reviewer YCyZ.

Following are some common questions and we have responded in each Reviewer's rebuttal:
1. model design in the response to LEpa and YCyZ.
2. novelty in the response to fTQV and YCyZ.
3. baseline models in the response to LEpa, fTQV and YCyZ.
4. experimental details in the response to LEpa, fTQV and YCyZ.
5. N-body MNIST dataset in the response to LEpa, and DF9s.
6. more discussions on references in the response to LEpa, fTQV, YCyZ and DF9s.
7. limitations on evaluation metrics, and future works in the response to fTQV and YCyZ.
8. limitations on incorporating physical constraints in the response to fTQV.
9. potential extension to video datasets in the response to LEpa and fTQV.

We are sincerely looking forward to further discussion.

Thanks,

Authors of Earthformer

---

### Meta-Review · Area_Chair_CBXE · 2022-08-27

**Recommendation:** Accept
**Confidence:** Certain

**Metareview:**

This work proposes a Transformer-based architecture using 3D blocks for spatio-temporal prediction, designed specifically for Earth system forecasting applications. The authors show that this method considerably outperforms other approaches both on two climate/weather forecasting tasks and on unrelated synthetic tasks. The reviewers agree that this is a strong submission, and it addresses an important area of problems too often neglected by our community. While one reviewer held a conflicting opinion to the other three, this review was completed 20 days after the deadline, and no response was made by this reviewer to the author rebuttal despite my requests to the reviewer. I believe that their concerns have been adequately addressed by the authors.

 I accordingly recommend acceptance of this paper.

**Award:**

No

---

### Decision · Program_Chairs · 2022-09-14

Accept